# Differences in the Clinical and Molecular Profiles of Subungual Melanoma and Acral Melanoma in Asian Patients

**DOI:** 10.3390/cancers15174417

**Published:** 2023-09-04

**Authors:** So-Young Ahn, Go-Eun Bae, Seung-Yeol Park, Min-Kyung Yeo

**Affiliations:** 1Department of Rehabilitation Medicine, Chungnam National University School of Medicine, Daejeon 35015, Republic of Korea; asyoung@cnuh.co.kr; 2Department of Pathology, Chungnam National University School of Medicine, Daejeon 35015, Republic of Korea; goeunbae@cnuh.co.kr; 3Department of Life Sciences, Pohang University of Science and Technology (POSTECH), Pohang 37673, Gyeongbuk, Republic of Korea

**Keywords:** subungual melanoma, acral melanoma, next-generation sequencing, Asian

## Abstract

**Simple Summary:**

Melanoma is a type of skin cancer that develops from melanocytes (the cells that give the skin its brown color). Subungual melanoma (nail melanoma) is a rare type of malignant melanoma that arises beneath the nails. Subungual melanoma has been categorized as a type of acral melanoma, which occurs on the hands and feet. Using a genetic study, we found that subungual melanoma showed molecular features that were different from those of acral melanoma. Subungal melanoma had a frequently mutated gene, named G protein subunit alpha Q. Patients with subungual melanoma show better survival than patients with acral melanoma. We suggest that subungual melanoma should be considered as a type of melanoma that is separate from acral melanoma. The genetic evaluation of subungual melanoma is important for advanced cancer patients, so that appropriate targeted therapy may be selected.

**Abstract:**

Subungual melanoma (SUM) is a rare type of malignant melanoma that arises beneath the nails. SUM is categorized as a type of acral melanoma (AM), which occurs on the hands and feet. SUM is an aggressive type of cutaneous melanoma that is most common among Asian patients. Recent studies reveal that SUM and AM might have different molecular characteristics. Treatment of melanoma relies on analysis of both clinical and molecular data. Therefore, the clinical and molecular characteristics of SUM need to be established, especially during metastasis. To define the mutation profiles of SUM and compare them with those of AM, we performed next-generation sequencing of primary and metastatic tumors of SUM and AM patients. Subungual location was a better independent prognostic factor than acral location for better overall survival (*p* = 0.001). Patients with SUM most commonly had the triple wild-type (75%) driven by *GNAQ* (58%) and *KIT* (25%) mutations, whereas patients with AM had *BRAF* (28.6%) and *RAF* (14.3%) molecular types of mutations. Single-nucleotide variations (SNVs) were more common in SUM than in AM, whereas copy number alterations (CNAs) were more common metastatic lesions of AM. Metastatic tumors in patients with SUM and AM showed increases in CNAs (43% and 80%, respectively), but not in SNVs. The number of CNAs increased during metastasis. When compared with AM, SUM has distinct clinical and molecular characteristics.

## 1. Introduction

Subungual melanoma (SUM) is categorized as a subtype of acral melanoma, which occurs on the hands and feet. SUM is a cutaneous malignant melanoma that arises from structures within the nail apparatus. Although SUM is a rare variant of melanoma in Caucasians (1–3%) [1], it is much more prevalent in Asian and African people with dark-pigmented skin (10–75% of melanoma cases) [2]. SUM is the most common cutaneous melanoma in Asian patients [3].

Recently, several studies showed that SUM and AM have different clinical and molecular characteristics. SUM and AM show different genetic signatures and different frequencies of mutated genes [4,5]. AM appears to have little or no relation to ultraviolet radiation (UVR); however, SUM has a UVR-related mutational signature [5,6]. AM is an aggressive type of melanoma with a poor prognosis; SUM is also thought to have a poor prognosis [2]. However, the 5-year survival rates for SUM patients range from 40% to 100%, depending on the clinical stage at the time of diagnosis. Some cases of SUM are diagnosed at an advanced stage, because it is often difficult to differentiate SUM from benign pigmented lesions of the nails [7,8]. Treatment options for SUM patients are controversial and include amputation or wide local excision of SUM tumors. This range of treatment options is due to the lack of comprehensive analysis of the clinicopathologic features of SUM and the survival rates of SUM patients, which makes clinical management difficult [9].

A few studies have been carried out to validate the molecular profiles of SUM. However, the number of patients enrolled in those previous studies have been limited. Some patients with SUM have not been evaluated separately from AM, and such studies validated only a few oncogenes, including BRAF mutations.

In the current study, we enrolled patients with SUM who had attended a single institution for 15 years and conducted next-generation sequencing (NGS) targeting 170 cancer-related genes, using primary and metastatic tumors of SUM and AM. Our hypothesis was that SUM and AM show distinct molecular characteristics in primary tumors and might have different genetic alteration during metastasis. The validation of the genetic changes in metastatic lesions is challenging, because targeted treatment and immunotherapy is indicated for advanced stages of melanomas. Treatment of melanoma relies on both clinical and molecular information to select appropriate surgical and therapeutic management. A better understanding of the distinct clinical and molecular characteristics of SUM is required to develop improved care pathways for melanoma patients.

## 2. Materials and Methods

### 2.1. Patients

Based on clinical photographs and medical records, 42 patients with SUM and 64 patients with AM were selected from the database of patients diagnosed with melanomas between 1 January 2008 and 31 December 2022 at the Chungnam National University Hospital, Daejeon, South Korea. The clinical data included age, sex, the location of the tumor, lymph node metastases, and distant metastases (Table 1). Survival curves for patients who died from melanoma were determined, based on data obtained from the medical records. Slides of biopsies or excised SUM and AM tumors were reviewed by two of the authors who are pathologists (M.-K.Y. and G.-E.B.). They collected the following data: the histopathologic subtype, the Breslow thickness of the tumors, ulceration, and the mitotic rate (mitotic count/mm^2^). The stages of the tumors were determined on the basis of the most recent classifications set out in 8th edition of the staging manual of the American Joint Committee on Cancer (AJCC) [10].

This retrospective study was approved by the Chungnam National University Hospital’s institutional review board (IRB file no. CNUH 2020-09-015), which waived the requirement for informed consent. All samples were provided by the Biobank of the Chungnam National University Hospital, which is a member of the Korea Biobank Network.

### 2.2. Tissue Samples and NGS

DNA/RNA was extracted from formalin-fixed paraffin-embedded (FFPE) tissues obtained from 16 patients with SUM and 7 patients with AM. Somatic alterations were analyzed quantitatively by the NGS targeting of 170 cancer-related genes (insufficient DNA/RNA was extracted from FFPE-tissue samples from the excluded patients). DNA and RNA were isolated from 20 µm sections of FFPE-tumor-tissue samples using a sterile 26-gauge needle and the RecoverAll™ Multi-Sample RNA/DNA Isolation Workflow (Ambion, Austin, TX, USA). Each tumor component was obtained by manual microdissection. DNA and RNA were extracted for library preparation. For each case, normal control tissue was dissected from an adjacent non-malignant region. DNA and RNA were quantified using a Qubit 2.0 fluorometer (Thermo Fisher Scientific, Waltham, MA, USA). Libraries were generated from 10 ng of DNA or RNA per sample, using the IonAmpliSeq™ Kit for Chef DL8, the Ion 540 Chef kit, and the Ion S5™ Chef system (all from Thermo Fisher Scientific).

Sequencing was performed using the Ion S5 sequencer and Ion 540 chips (Thermo Fisher Scientific). Sequence data analysis was performed using commercial pan-cancer Oncomine Comprehensive Assay version 3, Torrent Suite version 5.10.2, and Ion Reporter version 5.6 (Thermo Fisher Scientific). The Oncomine panel enabled analysis of variations in 170 genes, including copy number alterations (CNAs) in 47 genes and fusion drivers in 51 genes (Appendix A). Abbreviations of selected genes from the panel are described in Table 1. The workflow was created by adding a custom hotspots Browser Extensible Data file to report mutations of interest and a custom CNA baseline using the manufacturer’s default workflow, as previously described [11,12].

ANNOVAR software (http://www.openbioinformatics.org/annovar, accessed on 10 February 2023) was used for the functional annotation of the identified single-nucleotide variations (SNVs) to investigate their genomic locations and variations [13]. To eliminate error artifacts, sequence data were confirmed visually using the Integrative Genomics Viewer. This workflow identified SNVs and indels with a variant allele fraction as low as 1%. Based on the results of a feasibility study, the variant allele fraction threshold was set at 2%.

Somatic SNVs/indels that passed filtering for gain-of-function genes were considered gain-of-function if they occurred at predefined hotspot residues targeted by the Oncomine panel. Somatic variants of loss-of-function genes (oncogenes) were considered loss-of-function when deleterious (nonsense or frame-shifting) changes occurred at a predefined hotspot residue. Copy number analysis was performed using the copy number module within the previously mentioned Ion Reporter system workflow. If the copy numbers of target genes were ≥4, they were considered to be amplifications. Additionally, if the copy numbers of target genes were ≤1, they were considered to be deletions. Somatic CNAs were considered for potential actionability analysis when they were concordant with alteration (amplifications or deletions) predicted by Oncomine analysis. Somatic gene fusions were considered for actionability analysis when they represented known gene fusions listed in the Mitelman database (National Cancer Institute, Bethesda, MD, USA) or were determined by Oncomine analysis, or when they involved known 3′ or 5′ drivers with novel partners. These prioritized variants were associated with potential actionability using the Oncomine database. For each patient, the “most actionable” alteration was identified based on the following criteria: (i) variants referenced in Food and Drug Administration (FDA) drug labels; (ii) variants referenced in National Comprehensive Cancer Network (NCCN) treatment guidelines for the patient’s cancer type; (iii) variants referenced in an NCCN guideline for another cancer type; and (iv) variants referenced as inclusion criteria in a clinical trial. Actionable variants were identified by manual curation of FDA labels and NCCN guidelines, as well as by keyword searches and manual curation of clinical trial records in the TrialTrove database [11]. The genetic variants identified were interpreted and categorized as “pathogenic”, “likely pathogenic”, “variant of unknown significance”, “presumed benign,” or “benign”, based on clinical significance according to ClinVar-indexed variants (National Center for Biotechnology Information, USA) [12]. When assessing the mutation frequency of individual genes, “pathogenic” and “presumed pathogenic” were counted as mutations, while “benign” “presumed benign”, and “variants of unknown significance” were excluded.

### 2.3. Statistical Analysis

The Pearson χ2 test or Fisher’s exact test were used (as appropriate) to compare the baseline characteristics of the subgroups and to compare categorical variables. For univariate analysis, overall survival curves were generated using the Kaplan–Meier method and analyzed using the log-rank test. Multivariate survival analysis was performed using a Cox proportional hazards regression model in which age, stage, and subungual location were entered as covariates. *p* < 0.005 was considered significant. All statistical analyses were performed using SPSS version 26.0 for Windows (SPSS, Inc., Chicago, IL, USA).

## 3. Results

### 3.1. Characteristics of Patients with SUM

The baseline demographic and clinical characteristics of the 42 SUM patients are listed in Appendix A. All SUM patients (20 men and 22 women) were Asian (Korean), with a mean age of 55.7 (1–91) years. Twenty-nine (69.0%) SUM patients had fingernail involvement and 13 (31.0%) had toenail involvement. The mean size of the SUMs was 1.2 cm (0.2–4 cm). Twelve patients (28.6%) had in situ melanomas. In addition, 12 (28.6%) patients had tumors with a Breslow thickness <1 mm; four (9.5%) had tumors between 1 and 2 mm thick; five (11.9%) had tumors between 2 and 4 mm thick; and nine (21.4%) had tumors >4 mm thick. Regarding the stages of disease, 12 patients (28.6%) had stage 0 disease, 11 (26.2%) had stage I disease, 12 (28.6%) had stage II disease, six (14.3%) had stage III disease, and one (2.4%) had stage IV disease (based on the AJCC cancer-stage group definition). Eleven SUM patients (26.2%) had histological ulceration. The mean tumor mitotic rate/mm^2^ was 3 (0–22/mm^2^). Seven SUM patients (16.7%) had lymph node involvement and one (2.4%) had distant metastasis at the time of diagnosis. Five patients (11.9%) had a history of trauma at the site of melanoma. Twelve patients (28.6%) received chemotherapy, including targeted molecular agents; 11 (26.2%) received immunotherapy; and three (26.2%) received radiotherapy during follow-up after diagnosis of SUM.

### 3.2. Differences in the Clinicopathologic Characteristics of Patients with SUM or AM

Differences in the clinicopathologic characteristics of the tumors in 42 patients with SUM and 64 patients with AM, including in situ lesions, are listed in Table 2. Patients with SUM were younger (mean age, 55.7 years) than those with AM (67.3 years) (*p* = 0.009). The majority of primary SUMs developed on the nails of the hands (69%), in contrast to primary AMs, which mostly developed on the sole of the foot (81.3%) (*p* < 0.0001). The depth of SUM tumor invasion was shallower (the most frequent Breslow thickness was <1 mm (28.6%)) than that of AM (the most frequent Breslow thickness was >4 mm (29.7%)) (*p* = 0.037). The stages of SUM were earlier than those of AM, with advanced tumors (stages III-IV) reported less often for SUM (16.7%) than for AM (18.8%) (*p* = 0.016). More patients with SUM (11.9%) reported prior traumatic events; no patient with AM reported a prior traumatic event (*p* = 0.005). Loco-regional recurrence was less common in patients with SUM (4.8% for SUM patients compared with 17.2% for AM patients), but this difference was not statistically significant (*p* = 0.056). There were no significant differences related to sex, lymph node, or distant metastases at the time of surgery or during follow-up (*p* = 0.810, 0.944, 0.821, 0.145, and 0.622, respectively).

Differences in the clinicopathologic characteristics of the tumors in 30 patients with SUM and 59 patients with AM with invasive melanoma are listed in Table 3. The majority of primary SUMs developed on the nails of the hands (60%), in contrast to 16.9% of primary AMs (*p* < 0.0001). More patients with SUM (13.3%) reported prior traumatic events; no patient with AM reported a prior traumatic event (*p* = 0.004). Patients with SUM tended to be younger than those with AM (*p* = 0.054). There were no significant differences in sex, Breslow thickness, stage, lymph node metastases, or distant metastases when comparing SUM and AM and excluding in situ lesions (*p* = 0.936, 0.528, 0.745, 0.603, and 0.989, respectively).

### 3.3. Differences in Overall Survival of Patients with SUM or AM

Kaplan–Meier overall survival curves of patients with SUM and AM were stratified according to stage and Breslow thickness (*p* = 0.004 and *p* < 0.001, respectively) (Appendix A). Patients with advanced-stage disease and greater invasion depth had shorter overall survival. Univariate analysis of overall survival revealed that patients with SUM had a better prognosis than patients with AM (*p* = 0.001; Figure 1a). Patients with SUM had a better prognosis than patients with AM, including patients with only invasive melanomas (*p* = 0.029) (Figure 1b). The mean survival time of SUM patients was 6599 days (18 years), while the mean survival time of AM patients was 2744 days (7.5 years). The 5-year survival rate was 87.1% for SUM patients and 48.7% for AM patients.

A Cox proportional hazards regression model for multivariate analysis of prognostic factors, in which age (under 63 vs. over 63), stage (0–IV), and SUM vs. AM were entered as covariates, revealed that SUM and stage were independent prognostic factors for the overall survival of patients with SUM or AM (hazard ratio for AM = 2.936 and *p* = 0.016) (Table 4). Multivariate analysis identified SUM and lower stage of disease as prognostic factors for better overall survival (*p* = 0.016 and *p* = 0.038, respectively). A Cox proportional hazards regression model, excluding in situ lesions, was used for multivariate analysis of prognostic factors. This model also revealed that SUM and stage were independent prognostic factors for the overall survival of patients with SUM or AM with invasive lesions (hazard ratio for AM = 2.462 and *p* = 0.040) (Table 5).

### 3.4. Differences of Somatic Mutational Signatures of Patients with SUM or AM

Somatic alterations in 16 patients with primary and metastatic SUM and seven patients with metastatic AM were analyzed quantitatively by the NGS-targeting of 170 cancer-related genes according to molecular classification, origin of lesion, and stage of disease (Figure 2). Molecular testing for common melanoma driver genes, including *BRAF*, *NRAS*, *KRAS*, *HRAS*, *NF1*, and *KIT*, was available for this study.

Primary SUM most commonly presented with triple wild-type (negative for *BRAF*, *RAS*, *NF1*) in 75% of cases, which were driven by *GNAQ R183Q* (58%) and *KIT* (25%) mutations (Figure 2). The remaining primary SUM consisted of the *RAS* subtype (12.5%), the *BRAF* subtype (6.3%), and the *NF1* subtype (6.3%). Patients with primary SUM had *GNAQ R183Q* (80%), *KIT* including *K642E*, *V654A*, *N655K*, *D820Y* (31.3%), *NF1 S413* (6.3%), *KRAS G12D* (6.3%), *NRAS G13D* (6.3%), *BRAF V600E* (6.3%), *PIK3CA E542K* (6.3%), *CTNNB1 T41I* (6.3%), and *ATM A135* (6.3%) mutations (Figure 3a). SNVs (SUM4-SUM9) were detected in patients with early-stage (in situ or T1a) primary SUM. The disease-associated CNAs detected in the primary SUM were *KIT* (25%), *CDK4* (25%), *PDGFRA* (18.8%), *MDM2* (18.8%), *TERT* (12.5%), and *CCND2* (6.3%) amplifications (Figure 3b).

Somatic alterations of metastatic SUM and metastatic AM were compared. Metastatic SUM most commonly presented with the *KIT* mutation (71.4%), followed by *GNAQ R183Q* (28.6%), *PIK3CA E542K* (14.3%), *BRAF V600E* (14.3%), *NRAS G13D* (14.3%), and *CTNNB1 T41I* (14.3%) mutations (Figure 4a). Metastatic AM most commonly presented with *BRAF*, including *V600E* and *K601E* (28.6%) followed by *NRAS Q61K* (14.3%) mutations (Figure 4a). The disease-associated CNAs detected in the patients with metastatic SUM were *KIT* (42.3%), *TERT* (42.3%), *PDGFRA* (28.6%), *CDK4* (28.6%), *MDM2* (28.6%) amplifications, and *CDKN2A* (28.6%) and *CDKN2B* (33.3%) deletions (Figure 4b). The common disease-associated CNAs detected in the patients with metastatic AM were *CDKN2A* (57.1%) deletion, *BRAF* (42.9%) and *TERT* (42.9%) amplification, *CDKN2B* (28.6%) deletion, and *CDK6* (28.6%) and *MYC* (28.6%) amplification (Figure 4b). One metastatic AM patient harbored an *MKRN1-BRAF* translocation (14.2%).

### 3.5. Comparison of Somatic Mutations of Primary and Metastatic Melanomas of SUM and AM

To compare the molecular profiles of paired primary and metastatic SUM and AM tumors, we examined seven patients with SUM and five patients with AM (Figure 5). All metastatic tumors retained all of the genetic SNVs identified in the primary tumors. Four out of seven patients (57%) with SUM and one out of five patients (20%) with AM had identical mutational signatures in the primary and metastatic tumors. In the metastatic lesions, tumors gained additional CNAs. Patients with SUM gained *CDKN2A* (2/7 patients; 28.6%) and *CDKN2B* (2/7 patients; 28.6%) deletions and *BRAF* (1/7; 14.3%), *TERT* (1/7; 14.3%), and *NOTCH2* (1/7; 14.3%) amplifications. Metastatic tumors from patients with AM gained *CDKN2A* (2/5; 40%) and *PTEN* (1/5; 20%) deletions and *BRAF* (2/5; 40%), *TERT* (2/5; 40%), *CDK6* (1/5; 20%), *MYC* (1/5; 20%), *CCND1* (1/5; 20%), *FGF19* (1/5; 20%), *FGF3* (1/5; 20%), and *MDM2* (1/5; 20%) amplifications.

We validated cases of SUM and AM (SUM13 and AM2; Figure 4) with double lesions of metastatic melanomas; one case (AM2) gained CNAs in the second metastatic tumor and the other (SUM13) had the same mutational signature in the first and second metastatic tumor.

## 4. Discussion

The clinical characteristics of our patients with SUM were compared with the clinical characteristics of patients reported in previous studies (Table 6). The previous studies showed different gender predominance, depending on the patients’ race and location [14,15,16,17,18]. The male-to-female ratio in our study was 0.91, and slightly more women than men had SUM. The mean age of our SUM patients was 56 years, which was similar to that reported in the literature (47–66 years) [8,18,19,20]. Most previous studies reported SUM located on the first fingernail [4,7,16,17,18,19,21]. We found that SUM was most common in the nails of the upper extremity (69%). The disease stage and depth of invasion of SUM varied, with many studies reporting stage II (38.7%) [21], stage III (52%) [22], Clark level IV (53.4% and 70%) [7,20], and Clark level IV/V (79%) as the most common stages [23]. Lee et al. reported that the most common stage was stage 0 (in situ) in 63.2% of SUM patients, and their study examined mostly Caucasian and African-American patients [8]. Here, we found that the most common stage was stage 0 (in situ) in 28.6% of the SUM patients and stage II in 28.6% of the SUM patients.

Melanoma of the subungual and palmoplantar areas is categorized as AM or acral lentiginous melanoma. SUM is a rare type of melanoma in Caucasians, while SUM and AM are more common subtypes in Asians. SUM is a subtype of AM, which is both aggressive and one of the most common types of melanoma in Asians [3]. In Asian patients, SUM accounts for 10–17.5% of cutaneous melanoma cases [26,28] and 26–30% of AM cases [21,29]. SUM accounted for 14% of melanoma cases and 28% of AM cases in our study (Appendix A). Several papers have reported that SUM has pathologic features that are similar to those of AM, which is in contrast to other variants of melanoma, including the lentigo maligna type, the superficial spreading type, and the nodular type [30,31]. SUM and AM have histologic similarities with respect to radial growth patterns, lower local growth rates, high content of spindle cells, and advanced stages [21,30,31]. 

Despite these histologic similarities, different clinical features between SUM and AM have been reported. Patients with SUM were younger than those with AM, a finding consistent with our own findings [4,7]. We found that SUM develops more frequently on the hands, while AM develops more frequently on the foot. Patients with SUM reported a higher incidence of previous trauma to the lesions than those patients with AM. A previous study reported that ulcerations were more common in patients with SUM than they were in patients with AM, although other studies (including ours) showed no difference between SUM and AM in that regard [4,7,20]. Previous studies reported no difference between SUM and AM in the depths of Breslow thickness, the pathologic stages, or the Clark levels [4,7,21]. Holman et al. showed that patients with SUM had an increased frequency of metastases, but took longer to develop distant metastasis [4]. In our study, we found that the depth of tumor invasion was shallower and the tumor stage was earlier in patients with SUM than in patients with AM. Loco-regional recurrence was less common in patients with SUM than in patients with AM, due to including in situ lesions of SUM. The stages and the depths of SUM and AM did not differ when in situ lesions were excluded.

Patients with AM are considered to have a grave prognosis and to develop metastases more quickly than those with other subtypes of melanoma. Mejbel et al. reported that patients with SUM have a higher risk of death, but subungual location was not an independent prognostic factor [7]. Holman et al. and Jung et al. reported no significant difference in the survival of patients with AM or SUM [4,20]. Here, we found that the 5-year survival rates for patients with SUM was 87.1%. Subungual rather than acral location was an independent prognostic factor for better overall survival. We hypothesized that the poor prognosis of patients with SUM, as reported in prior studies, might be due to late presentation or late implementation of clinical management. Usually, SUM presents as a brown-black discoloration of the nail bed, and it is commonly misdiagnosed as striate melanonychia and onychomycosis. Prior studies commonly reported patients with SUM at an advanced stage [20,21,22,23]. SUM tumors were often ulcerated and deep, resulting in 5-year survival rates of approximately 40% [19,20]. Mejbel et al. enrolled mostly advanced staged (stage IV in 70%) patients with SUM and reported 5-year overall survival rates of 40% [7]. In the present study, most patients were diagnosed early, as stage 0 or stage II (26.2%), and multivariate analysis adjusted for stage and age showed that patients with SUM had better overall survival than patients with AM. SUM patients, excluding in situ cases, also showed better overall survival than patients with AM. Lee et al. enrolled 63.2% of patients with in situ melanoma, and all of those patients with SUM survived during the 5-year follow-up [8]. They also found that early diagnosis was a good prognostic feature for SUM [8].

Recently, melanomas were classified into different subtypes according to their etiological relationship to sun exposure, mutational signatures, anatomic sites, and epidemiology. SUM and AM showed different molecular characteristics related to cumulative solar damage [4,5]. SUM had a mutational signature related to UVR, whereas AM appeared to have little or no relation to UVR [5,6]. Different molecular pathogenesis to UVR may affect clinical and molecular characteristics of SUM that are distinct from those of AM. The previous somatic mutations of SUM vary the *GNAQ* (0–25%), *KIT* (0–50%), *NF1* (0–50%), *BRAF* (0–43%), and *NRAS* (0–31%) mutations that have been reported [4,5,18,25,27,32,33,34,35,36,37,38,39,40] (Table 6). We found that most patients with SUM had triple wild-type disease driven by *GNAQ R183* and *KIT* mutations, whereas most patients with AM showed *BRAF* (including *V600* and *K601*) and *RAS (NRAS Q61*, *NRAS G13*, and *KRAS G12*) molecular types. Deletion of the cell cycle pathway gene *CDKN2A/B* (p16) occurred more frequently in patients with AM than in patients with SUM. Patients with SUM had more SNVs, but CNAs were more common in patients with AM. Haugh et al. described the higher frequency of CNAs of *CDK4* and *CCND1*, and few *BRAF* mutations with SUM [34]. Complex gene rearrangements in melanomas may be potential molecular therapeutic targets in melanoma patients [5]. The inconstant mutational profiles of SUM in previous studies require further investigation.

This is the first study to validate the molecular characteristics of matched primary and corresponding metastatic tumors from patients with SUM and to compare them with those of patients with AM. All of the metastatic tumors retained all genetic SNVs found in the primary tumors, and half of the patients with SUM and one-fifth of the patients with AM showed identical mutational signatures of primary and metastatic tumors. Interestingly, metastatic tumors gained additional CNAs, but not SNVs, and the number of CNAs increased in metastatic tumors. The metastatic tumors in AM showed more frequent acquisition of CNVs, but the mutational signatures of SUM and AM were similar. *CDKN2A* and *CKDN2B* deletions and *BRAF* and *TERT* amplifications were common in the metastatic tumors of SUM and AM. 

Turajlic et al. detected a wide ranged genetic heterogeneity (11–84%) in both the primary and metastatic tumors of AM [41]. Manca et al. used target panel NGS sequencing to show a high level of concordance between the mutational patterns of primary and metastatic cutaneous melanoma, reporting that the consistency of pathogenic mutations was 76% [42]. Herein, our panel NGS sequencing revealed mutational consistency in 57% of SUM primary and matched metastatic tumors and 25% of AM primary and matched metastatic tumors. Sanborn et al. conducted whole-exome sequencing of cutaneous melanomas and reported that genetically distinct cell populations in the primary tumor metastasized to different anatomic sites in 6 out of 8 cases, indicating a late-evolving trait of metastases [43]. Discrepancies between these prior studies and our study may be due to (i) different NGS platforms and interpretation of mutations, (ii) validation of different subtypes of melanoma, and (iii) studies involving different races. Turajlic et al. described highly similar levels of SNVs and CNAs in primary and metastatic tumors, and they suggested that metastatic tumor clones stem from non-modal sub-clones of the primary melanoma that emerged at a late stage of development of the primary melanoma, or that metastatic clones harbored specific mutations in addition to the mutations found in the modal population of the primary melanoma [41].

This study had several limitations, including the small sample size, the retrospective design, and the inclusion of patients from a single institution. The number of tissue samples enrolled were limited, due to a high failure rate of sequencing (more than 60%). Melanin pigment, decalcification during the sample processing from nails, and scattered tumor cells in the early lesion caused poor DNA quality. Patients with AM were only selected if they had both primary and metastatic tumor samples. The molecular profiles of primary SUM could not be compared equally with the results of AM, due to selection bias. However, our data highlight the distinct clinicopathologic features of SUM, new data regarding overall survival, and molecular data from Asian patients with SUM. Early detection of SUM results in a good prognosis for patients with SUM. Therefore, we propose that clinical management of SUM melanoma should be revalidated, based on race. A deeper understanding of the molecular and immunologic characteristics of SUM is required to select appropriate clinical treatment strategies, including targeted molecular agents.

## 5. Conclusions

Herein, we reported differences in the clinicopathologic and mutation profiles of SUM and AM in Asian patients. SUM was related to favorable clinical factors and showed better overall survival. The molecular profiles of SUM (mostly triple wild-type driven by the *GNAQ R183* and *KIT* mutations) were different from those of AM. SNVs were more common in metastatic lesions of SUM, whereas CNAs were more common in metastatic lesions of AM. Patients with SUM may require different therapies, depending on the distinct anatomical locations and molecular characteristics of the tumors. We suggest that SUM should not be categorized along with AMs. Rather, it should be considered as a separate entity.

## Figures and Tables

**Figure 1 cancers-15-04417-f001:**
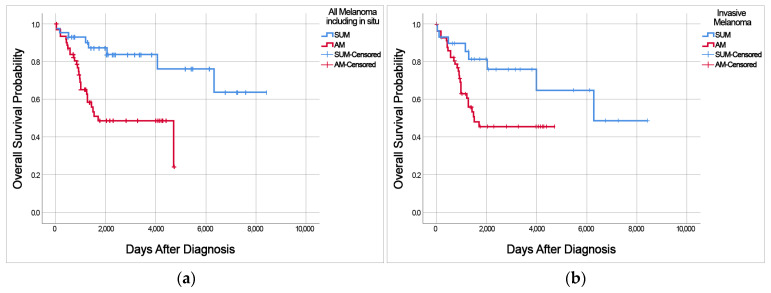
Kaplan–Meier curves showing overall survival of patients with SUM and AM. (**a**) All melanoma patients with SUM and AM, including in situ melanoma; (**b**) patients with invasive melanoma.

**Figure 2 cancers-15-04417-f002:**
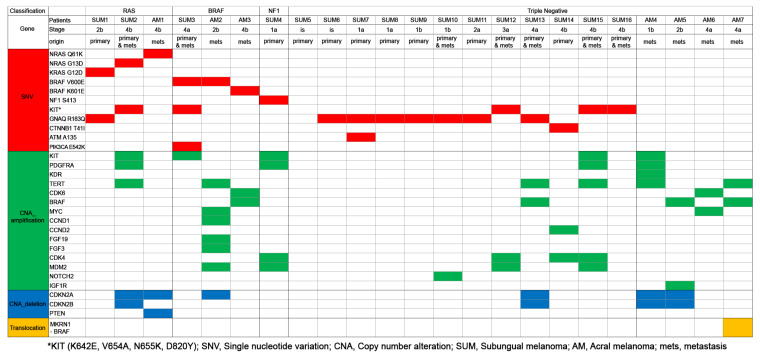
Somatic mutational signatures of subungual and acral melanomas. *, amino acid changes of the *KIT* gene. Red bar, single nucleotide variation. Green bar, copy number amplification; Blue bar, copy number deletion. Yellow bar, Translocation.

**Figure 3 cancers-15-04417-f003:**
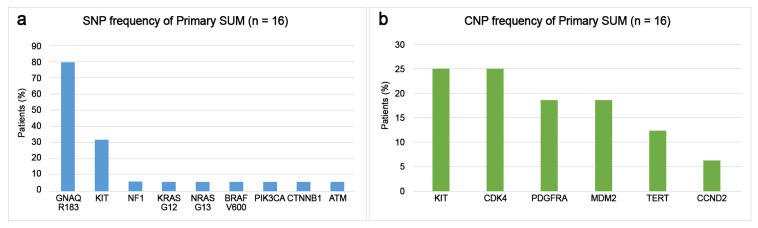
Frequency of alterations in mutated genes of patients of subungual melanoma (SUM): (**a**) single-nucleotide polymorphism (SNP) frequencies and (**b**) copy number alteration (CNA) frequencies.

**Figure 4 cancers-15-04417-f004:**
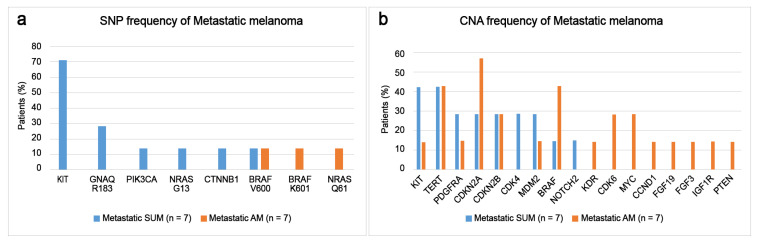
Changes in somatic mutation alterations of primary and metastatic subungual and acral melanomas: (**a**) single-nucleotide polymorphism (SNP) frequencies and (**b**) copy number alteration (CNA) frequencies of metastatic melanomas.

**Figure 5 cancers-15-04417-f005:**
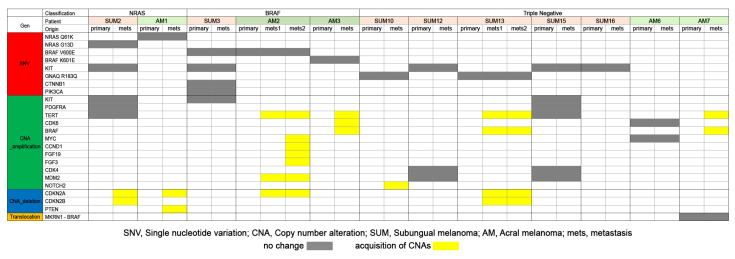
Somatic mutational alteration comparison of primary and metastatic tumors of subungual and acral melanoma. Gray bar, no change of the mutation. Yellow bar, acquisition of copy number alteration in the metastatic tumor.

**Table 1 cancers-15-04417-t001:** List of selected genes of NGS targeted cancer-related genes.

Abbreviation	Gene Name
*GNAQ*	G protein subunit alpha Q
*KIT*	KIT proto-oncogene receptor tyosine kinase
*BRAF*	V-raf murine sarcoma viral oncogene homolog B1
*RAS*	Rat sarcoma virus
*NRAS*	Neuroblastoma RAS viral oncogene homolog
*KRAS*	Kirsten rat sarcoma virus
*HRAS*	Harvey rat sarcoma viral oncogene homolog
*NF1*	Neurofibromin 1
*CTNNB1*	Catenin beta-1
*ATM*	Ataxia telangiectasia mutated
*CDK4*	Cyclin-dependent kinase inhibitor 4
*MDM2*	Murine double minute 2
*PDGFRA*	Platelet derived growth factor receptor alpha
*CCND2*	Cyclin D2
*PIK3CA*	Phosphatidylinositol-4,5-bisphosphate 3-kinase catalytic subunit alpha
*TERT*	Telomerase reverse transcriptase
*CDKN2A*	Cyclin-dependent kinase inhibitor 2A
*CDKN2B*	Cyclin-dependent kinase inhibitor 2B
*MYC*	Myelocytomatosis oncogene
*MKRN1*	Makorin ring finger protein 1
*NOTCH2*	Neurogenic locus notch homolog protein 2
*PTEN*	Phosphatase and tensin homolog
*FGF19*	Fibroblast growth factor 19
*FGF3*	Fibroblast growth factor 3

**Table 2 cancers-15-04417-t002:** Differences in the clinic-pathologic features of subungual (n = 42) and acral (n = 64) melanomas including in situ lesions.

	No. (%)	Subungual	Acral	*p*
Sex				0.810
Male	52 (49.1)	20 (47.6)	32 (50.0)	
Female	54 (50.9)	22 (52.4)	32 (50.0)	
Age (years)				0.009
≤63	49 (46.2)	26 (61.9)	23 (35.9)	
>63	57 (53.8)	16 (38.1)	41 (64.1)	
Tumor location				0.000
Upper	41 (38.7)	29 (69.0)	12 (18.8)	
Lower	65 (61.3)	13 (31.0)	52 (81.3)	
Invasion of the dermis				0.004
In situ melanoma	17 (16.0)	12 (28.6)	5 (7.8)	
Invasive melanoma	89 (84.0)	30 (71.4)	59 (92.2)	
Breslow thickness				0.037
In situ	17 (16.0)	12 (28.6)	5 (7.8)	
<1 mm	27 (25.5)	12 (28.6)	15 (23.4)	
1.01–2.00 mm	15 (14.2)	4 (9.5)	11 (17.2)	
2.01–4.00 mm	19 (17.9)	5 (11.9)	14 (21.9)	
>4.00 mm	28 (26.4)	9 (21.4)	19 (29.7)	
Lymph node metastasis at the time of diagnosis				0.944
Absent	88 (83.0)	35 (83.3)	53 (82.8)	
Present	18 (17.0)	7 (16.7)	11 (17.2)	
Distant metastasis at the time of diagnosis				0.821
Absent	103 (97.2)	41 (97.6)	62 (96.9)	
Present	3 (2.8)	1 (2.4)	2 (3.1)	
Stage group				0.016
0	17 (16.0)	12 (28.6)	5 (7.8)	
I–II	70 (66.0)	23 (54.8)	47 (73.4)	
III–IV	19 (17.9)	7 (16.7)	12 (18.8)	
Immunotherapy				0.696
Not carried out	76 (71.7)	31 (73.8)	45 (70.3)	
Carried out	30 (28.3)	11 (26.2)	19 (29.7)	
Trauma				0.005
Absent	101 (95.3)	37 (88.1)	64 (100.0)	
Present	5 (4.7)	5 (11.9)	0 (0.0)	
Ulceration				0.226
Absent	71 (67.0)	31 (73.8)	40 (62.5)	
Present	35 (33.3)	11 (26.2)	24 (37.5)	
Loco-regional recurrence				0.056
Absent	93 (87.7)	40 (95.2)	53 (82.8)	
Present	13 (12.3)	2 (4.8)	11 (17.2)	
Distant metastasis during the follow-up				0.622
Absent	78 (73.6)	32 (76.2)	46 (71.9)	
Present	28 (26.4)	10 (23.8)	18 (21.6)	

**Table 3 cancers-15-04417-t003:** Differences in the clinic-pathologic features of subungual (n = 30) and acral (n = 59) invasive melanomas.

	No. (%)	Subungual	Acral	*p*
Sex				0.936
Male	41 (46.1)	14 (46.7)	27 (45.8)	
Female	48 (53.9)	16 (53.3)	32 (54.2)	
Age (years)				0.054
≤63	35 (39.3)	16 (53.3)	19 (32.2)	
>63	54 (30.7)	14 (46.7)	40 (67.8)	
Tumor location				0.000
Upper	28 (31.5)	18 (60.0)	10 (16.9)	
Lower	61 (68.5)	12 (40.0)	49 (83.1)	
Breslow thickness				0.528
<1 mm	27 (30.3)	12 (40.0)	15 (25.4)	
1.01–2.00 mm	15 (16.9)	4 (13.3)	11 (18.6)	
2.01–4.00 mm	19 (21.3)	5 (16.7)	14 (23.7)	
>4.00 mm	39 (31.5)	9 (30.0)	19 (32.2)	
Lymph node metastasis at the time of diagnosis				0.603
Absent	71 (79.8)	23 (76.7)	48 (81.4)	
Present	18 (20.2)	7 (23.3)	11 (18.6)	
Distant metastasis at the time of diagnosis				0.989
Absent	86 (96.6)	29 (96.7)	57 (96.6)	
Present	3 (3.4)	1 (3.3)	2 (3.4)	
Stage group				0.745
I–II	70 (78.7)	23 (76.7)	47 (79.7)	
III–IV	19 (21.3)	7 (23.3)	12 (20.3)	
Immunotherapy				0.674
Not carried out	59 (66.3)	19 (63.3)	40 (67.8)	
Carried out	30 (33.7)	11 (36.7)	19 (32.2)	
Trauma				0.004
Absent	85 (95.5)	26 (86.7)	59 (100.0)	
Present	4 (4.5)	4 (13.3)	0 (0.0)	
Ulceration				0.714
Absent	54 (60.7)	19 (63.3)	35 (59.3)	
Present	35 (39.3)	11 (36.7)	24 (40.7)	
Loco-regional recurrence				0.179
Absent	77 (86.5)	28 (93.3)	49 (83.1)	
Present	12 (13.5)	2 (6.7)	10 (16.9)	
Distant metastasis during the follow-up				0.786
Absent	61 (68.5)	20 (66.7)	41 (69.5)	
Present	28 (31.5)	10 (33.3)	18 (30.5)	

**Table 4 cancers-15-04417-t004:** Multivariate analysis results for overall survival including in situ melanomas.

Melanoma Including In Situ Lesions	*p*	HR	95% CI
Subungual vs. Acral	0.016	2.936	1.225–7.036
Age (under 63 years vs. over 63 years)	0.261	1.529	0.730–3.205
Stage 0	0.038		
Stage I	0.408	2.474	0.289–21.177
Stage II	0.075	6.592	0.89–52.403
Stage III	0.039	9.331	1.119–77.801
Stage IV	0.167	7.328	0.436–123.133

HR, hazard ratio; CI, confidence index.

**Table 5 cancers-15-04417-t005:** Multivariate analysis results for overall survival with invasive melanomas.

Melanoma with Invasive Lesions	*p*	HR	95% CI
Subungual vs. Acral	0.040	2.462	1.044–5.806
Age (under 63 years vs. over 63 years)	0.279	1.504	0.718–3.151
Stage I	0.073		
Stage II	0.030	2.628	1.099–6.282
Stage III	0.013	3.723	1.325–10.459
Stage IV	0.328	2.909	0.342–24.716

HR, hazard ratio; CI, confidence index.

**Table 6 cancers-15-04417-t006:** Summary of clinical and mutational profiles of SUM in the literature.

Clinical and Molecular Profiles	Subungual Melanoma
Sex	Male predominance [4,7,14,15,17,21,24] Female predominance [8,14,15,16,25].
Mean age (years)	47–66 years [4,7,8,14,15,16,17,18,21].
The most common tumor location	Fingernail (17–63%) [4,7,14,16,17,18,19,21,25] Toenail (26.3%, 68%, and 78%) [15,24]
The most common Clark level	Clark level IV (37%, 53%, and 70%) [7,15,18,21]
The most common Breslow thickness	≤1 mm (63.2%) [8] 1.01–4 mm (34–50%) [7,18,21,24] >4 mm (31%, 34.9%, 50%) [14,20,25]
The most common stage group	0 (in situ) (63.2%) [8]. I (83%) [17] II (32.6–53.7%) [14,15,20] III (32% and 52%) [21,24]
Lymph node metastasis at the time of diagnosis	9–62.3% [15,16,17,18,21,24]
Distant metastasis at the time of diagnosis	9.3% and 42.6% [18,20]
Trauma	15.8% and 48% [4,8]
Ulceration	47–76% [7,14,15,16,17,20,21,24]
5-year disease free survival	40% and 57% [7,18]
5-year survival	74–97% [15,17,18,26]
Overall mortality	17%, 31%, and 46% [17,18,21]
*BRAF V600E* mutation	0–40% [4,5,14,16,27]
*NRAS* mutation	0–31% [4,14,16,27]
*KRAS* mutation	6.5% and 11% [4,27]
*NF1* mutation	0–50% [4,27]
*C-KIT* mutation	11.1%, 13%, and 16% [4,14,16,27]
*GNAQ* mutation	0–25% [27]

## Data Availability

The raw data were generated at CNUH and derived data supporting the findings of this study are available from the corresponding author on request.

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
