# Peer review of "Differences in the Clinical and Molecular Profiles of Subungual Melanoma and Acral Melanoma in Asian Patients"

_cancers, 2023, doi:10.3390/cancers15174417_

Round 1

Reviewer 1 Report

To define the mutation profiles of SUM and compare them with those of AM, This study performed next-generation sequencing of primary and metastatic tumors with SUM and AM. The idea of the paper is interesting however it needs substantial improvement.  Below are the comments that need to be addressed

1.     The results added in the abstract are generic, add statistical results

2.     The technical content is good however the motivation of the paper is not clear. Why is your proposal needed? What are the challenges involved? What solutions already exist for the problem you want to solve? What are their limitations and drawbacks? Author may add few subsections at the end of the introduction section to address all these concerns.

3.     May add some introductory lines between heading and sub-heading like 2 and 2.1; 3 and 3.1 etc

4.     Related work section is missing, add some latest related research and also add a table of comparison to identify the gaps in existing literature to provide the state-of-the-art

5.     Sentences are too long, try to reduce the length of sentences for better readability

6.     Extensive proofread is required

7.     Figures quality need to be improved especially figure 3 and 4

8.     Compare your findings with some existing solutions in the discussion section and present it in a tabular or graphical form

9.     Add a Limitation section as a subsection in discussion section

10.  References are not enough, may add few more references

11.  Conclusion is very short, map it with abstract

 Moderate editing of English language required

Author Response

Reply to the Reviewer 1

Comments to Author: To define the mutation profiles of SUM and compare them with those of AM. This study performed next-generation sequencing of primary and metastatic tumors with SUM and AM. The idea of the paper is interesting however it needs substantial improvement.  Below are the comments that need to be addressed.

  1. The results added in the abstract are generic, add statistical results

Response: We revised abstract and added statistical results.

  1. The technical content is good however the motivation of the paper is not clear. Why is your proposal needed? What are the challenges involved? What solutions already exist for the problem you want to solve? What are their limitations and drawbacks? Author may add few subsections at the end of the introduction section to address all these concerns.

Response: We appreciated your careful reviews and comments to the paper. We added a few subsections at the end of the introduction and addressed the motivation of this study (line 50-62). We thought the molecular results from the previous studies related to SUM were limited due to a small number of patients enrolled and a few targeting genes they applied before. We supposed that subungual and acral melanoma has its distinct molecular characteristic in the primary tumors and also guessed subungual and acral melanoma might have different genetic alteration during the metastasis. The validation of genetic changes of metastatic melanomas can be challenging because targeted treatment and immunotherapy are indicated for advanced stages of melanomas. We added a few subsections in the discussion and addressed the limitations and drawbacks of this study (line 378-384).

  1. May add some introductory lines between heading and sub-heading like 2 and 2.1; 3 and 3.1 etc

Response: We added introductory line between headings and sub-headings in the 2. Materials and 3. Results sections.

  1. 4.     Related work section is missing, add some latest related research and also add a table of comparison to identify the gaps in existing literature to provide the state-of-the-art

    Response: We added more references of related research (ref. 30-43). We added a table (Table 6) of summary for clinical and mutational profiles of subungual melanoma in the literature and compared with our study (line 277-288).

  1. Sentences are too long, try to reduce the length of sentences for better readability

     Response: We appreciated your comments to the paper. We performed an editing service and revised the document. We tried to revise and correct sentences for better readability.

  1. Extensive proofread is required

    Response: We revised the document with an editing service and tried to remove and correct errors in the document.

  1. Figures quality need to be improved especially figure 3 and 4

Response: We tried to improve figure qualities of Figure 3 and 4.

  1. Compare your findings with some existing solutions in the discussion section and present it in a tabular or graphical form

   Response: We added more related research and added a table of summary of clinical and mutational profiles of subungual melanoma in the literature (Table 6). W compared our findings with existing results in the discussion section (line 278-288 and 340-346).

  1. Add a Limitation section as a subsection in discussion section

Response: We added a few subsections in the discussion and addressed the limitation of this study (line 378-384). This study had several limitations, including the small sample size, a retrospective design, and inclusion of patients from a single institution. The number of tissue samples enrolled were limited due to a high failure rate of sequencing (more than 60%); melanin pigment, decalcification during the sample processing from nails, and scattered tumor cells in the early lesion caused poor DNA quality. Because, patients with AM were only selected having both primary and metastatic tumor samples, the molecular profiles of primary SUM could not be equally compared to the results of AM.

  1. References are not enough, may add few more references

Response: We appreciated your reviews and comments to the paper. We added more references of related research (ref. 30-43).

  1. Conclusion is very short, map it with abstract

Response: We revised the conclusion and added contents to the conclusion.

- Moderate editing of English language required

Response: We performed an editing service and revised the document again. We tried to revise and correct sentences for better readability.

Reviewer 2 Report

In this manuscript, the authors studied the clinical and genetic mutation profiles of sublingual and acral melanoma in Asian patients. The study can be a useful resource for the melanoma research community. However, there are numerous grammar issues and careless writing throughout the manuscript. Please thoroughly revise.

Please change triple-negative wild-type to triple wild-type.

Line 108. There is NO hotspot loss-of-function mutation for tumor suppressors

Line 112. If copy number was <1, then the only option would be 0, or homozygous deletion. Were the authors only referring homozygous deletion in this situation?

Figure 2. For BRAF, please separate V600 mutations with other mutations; similarly, for NRAS, separate G12/Q61 mutations and other mutations.

For GNAQ separate R183/Q209 mutations and other mutations. It is unusual to have GNAQ mutations at such high frequency. If they are all pathogenetic mutations, such as R183/Q209 mutations, it would be very noteworthy.

There are numerous grammar issues and careless writing throughout the manuscript. Please thoroughly revise.

Author Response

Reply to the Reviewer 2

In this manuscript, the authors studied the clinical and genetic mutation profiles of sublingual and acral melanoma in Asian patients. The study can be a useful resource for the melanoma research community. However, there are numerous grammar issues and careless writing throughout the manuscript. Please thoroughly revise.

Please change triple-negative wild-type to triple wild-type.
Response: We changed expressions “triple-negative wild-type” to “triple wild-type”.

 Line 108. There is NO hotspot loss-of-function mutation for tumor suppressors
Response: We appreciated your reviews and comments to the paper. We revised the sentences to avoid the misinterpretation of the somatic mutation calling. (line 113).

Line 112. If copy number was <1, then the only option would be 0, or homozygous deletion. Were the authors only referring homozygous deletion in this situation?
Response: We corrected as “copy number was ≤1” and our study included homozygous and heterozygous deletions (line 118).

Figure 2. For BRAF, please separate V600 mutations with other mutations; similarly, for NRAS, separate G12/Q61 mutations and other mutations.
Response: We separated BRAF V600 mutations with other mutations, and NRAS G12/Q61 mutations and other mutations. We revised the description of BRAF, NRAS, KRAS mutations separately in the results and discussion sections and also in the Figure 2-5.

For GNAQ separate R183/Q209 mutations and other mutations. It is unusual to have GNAQ mutations at such high frequency. If they are all pathogenetic mutations, such as R183/Q209 mutations, it would be very noteworthy.

Response: In our study, subungual melanoma patients all showed pathogenic GNAQ R183Q mutation. Previous studies did not describe GNAQ mutation (line 342-343, ref. 30) or they included a small number of subungual melanoma in the cases (line 342-343, ref.30-43). In recent studies, Holman et al. described KIT mutations (triple wild type) were more common in subungual melanoma than non-nail acral melanoma (ref. 42). We suggested a high GNAQ mutation can be a characteristics of subungual melanoma as a triple wild type subtype in Asian patients.

There are numerous grammar issues and careless writing throughout the manuscript. Please thoroughly revise.

Response: We appreciated your reviews and comments to the paper. We performed an editing service and revised the document. We tried to remove and correct errors in the document.

Reviewer 3 Report

The MS “Differences in the clinical and molecular profiles of subungual  melanoma and acral melanoma in Asian patients” (by So Young Ahn et al.) describes the two less usual types of melanoma: acral melanoma (AM) and subungual melanoma SUM). The paper is mostly a clinical observation describing clinicopathological differences between the two tumor types. The addition of NGS data about mutations improves the scientific soundness of the paper. The authors should follow the minor concerns given below. Overall, the paper is well written with a good statistical analysis. Notably, the frequency of BRAF mutation is much less than found in skin melanomas appearing on sun exposed sites.

Minor concerns:

Figure 3a,b: Only data from SUM patients are shown while in the legend the acral melanomas are also mentioned. Please explain/amend.

The addition of a paragraph of abbreviations summary at the beginning (after abstract) would help, there are many abbreviations through the text, even though they are explained at first appearance.

Line 379: The However, ……..? please correct

Author Response

Reply to the Reviewer 3

The MS “Differences in the clinical and molecular profiles of subungual melanoma and acral melanoma in Asian patients” (by So Young Ahn et al.) describes the two less usual types of melanoma: acral melanoma (AM) and subungual melanoma SUM). The paper is mostly a clinical observation describing clinicopathological differences between the two tumor types. The addition of NGS data about mutations improves the scientific soundness of the paper. The authors should follow the minor concerns given below. Overall, the paper is well written with a good statistical analysis. Notably, the frequency of BRAF mutation is much less than found in skin melanomas appearing on sun exposed sites.

 Minor concerns:

Figure 3a,b: Only data from SUM patients are shown while in the legend the acral melanomas are also mentioned. Please explain/amend.

Response: We corrected typos in the legend of figure 3. Frequency of alterations in mutated genes of patients of subungual melanoma were only shown in Figure 3 (Page 8).

The addition of a paragraph of abbreviations summary at the beginning (after abstract) would help, there are many abbreviations through the text, even though they are explained at first appearance.

Response: We added a table of list of selected genes of NGS targeting cancer-related genes (Page 3, Table 1). The abbreviation summary of selected genes in the text was included in the Table 1.  

Line 379: The However, ……..? please correct

Response: We appreciated your comments and corrected errors in the discussion section (line 385).

Round 2

Reviewer 1 Report

The author have addressed all my comments, I have no more comments. The paper is now acceptable for publication

Author Response

The author have addressed all my comments, I have no more comments. The paper is now acceptable for publication

Response: We appreciated your careful reviews and comments to the paper. We could have a chance to improve our paper.